# Understanding the Conundrum of Pancreatic Cancer in the Omics Sciences Era

**DOI:** 10.3390/ijms25147623

**Published:** 2024-07-11

**Authors:** Alberto Nicoletti, Mattia Paratore, Federica Vitale, Marcantonio Negri, Giuseppe Quero, Giorgio Esposto, Irene Mignini, Sergio Alfieri, Antonio Gasbarrini, Maria Assunta Zocco, Lorenzo Zileri Dal Verme

**Affiliations:** 1CEMAD Centro Malattie dell’Apparato Digerente, Medicina Interna e Gastroenterologia, Dipartimento di Medicina e Chirurgia Traslazionale, Università Cattolica del Sacro Cuore, Fondazione Policlinico Universitario “A. Gemelli” IRCCS, 00168 Rome, Italy; alberto.nicoletti@policlinicogemelli.it (A.N.); mattia.paratore@guest.policlinicogemelli.it (M.P.); fede.vitale1996@gmail.com (F.V.); marcantonio.negri@policlinicogemelli.it (M.N.); giorgio.esposto@guest.policlinicogemelli.it (G.E.); irene.mignini@guest.policlinicogemelli.it (I.M.); antonio.gasbarrini@unicatt.it (A.G.); lorenzo.zileridalverme@policlinicogemelli.it (L.Z.D.V.); 2Centro Pancreas, Chirurgia Digestiva, Dipartimento di Medicina e Chirurgia Traslazionale, Università Cattolica del Sacro Cuore, Fondazione Policlinico Universitario “A. Gemelli” IRCCS, 00168 Rome, Italy; giuseppe.quero@policlinicogemelli.it (G.Q.); sergio.alfieri@unicatt.it (S.A.)

**Keywords:** pancreatic cancer, omics sciences, metabolomics, proteomics, genomics, radiomics, translational research, precision medicine, personalized medicine

## Abstract

Pancreatic cancer (PC) is an increasing cause of cancer-related death, with a dismal prognosis caused by its aggressive biology, the lack of clinical symptoms in the early phases of the disease, and the inefficacy of treatments. PC is characterized by a complex tumor microenvironment. The interaction of its cellular components plays a crucial role in tumor development and progression, contributing to the alteration of metabolism and cellular hyperproliferation, as well as to metastatic evolution and abnormal tumor-associated immunity. Furthermore, in response to intrinsic oncogenic alterations and the influence of the tumor microenvironment, cancer cells undergo a complex oncogene-directed metabolic reprogramming that includes changes in glucose utilization, lipid and amino acid metabolism, redox balance, and activation of recycling and scavenging pathways. The advent of omics sciences is revolutionizing the comprehension of the pathogenetic conundrum of pancreatic carcinogenesis. In particular, metabolomics and genomics has led to a more precise classification of PC into subtypes that show different biological behaviors and responses to treatments. The identification of molecular targets through the pharmacogenomic approach may help to personalize treatments. Novel specific biomarkers have been discovered using proteomics and metabolomics analyses. Radiomics allows for an earlier diagnosis through the computational analysis of imaging. However, the complexity, high expertise required, and costs of the omics approach are the main limitations for its use in clinical practice at present. In addition, the studies of extracellular vesicles (EVs), the use of organoids, the understanding of host–microbiota interactions, and more recently the advent of artificial intelligence are helping to make further steps towards precision and personalized medicine. This present review summarizes the main evidence for the application of omics sciences to the study of PC and the identification of future perspectives.

## 1. Introduction

Pancreatic cancer (PC) is one of the most lethal malignancies. Despite its incidence being relatively low, accounting for about 3% of all cancers, it is a significantly higher (7%) cause of cancer-related death in the United States [1]. The aggressive biology of PC and the absence of clinical symptoms in the early phase of the disease delay the diagnosis until the disease is locally advanced or metastatic, when surgery, the only curative treatment, is inapplicable [2]. Indeed, the 5-year survival rate of patients treated with resection as part of their multimodal therapy reaches 20%. However, the prognosis remains dismal when patients are treated with chemotherapy alone or in association with radiation therapy for advanced disease [3].

Furthermore, the incidence of PC is estimated to increase until 2030, when it may become the second cause of cancer-related death [1]. In Western countries, this rise in incidence is due to the increasing prevalence of the risk factors for its development (e.g., smoking, alcohol abuse, fatty diet, obesity, and diabetes mellitus) [4,5].

Despite the efforts of researchers and improvements in surgical technique, the dismal prognosis of PC has not increased significantly in recent years.

PC is characterized by a complex tumor microenvironment consisting of neoplastic cells, endothelial vascular cells, immune cells, fibroblasts and myofibroblasts, stellate cells, and extracellular matrix. All of these components and their interactions play a crucial role in tumor development and progression, contributing to the alteration of metabolism and cellular hyperproliferation, as well as to metastatic evolution and abnormal tumor-associated immunity [6].

The advent of omics sciences is revolutionizing the comprehension of the pathogenetic conundrum of pancreatic carcinogenesis. For example, several authors have identified distinct subtypes of pancreatic ductal adenocarcinoma (PDAC) using molecular profiling-based genomic and transcriptomic techniques. Firstly, Collison et al. described three PDAC subtypes: the classical, the quasi-mesenchymal, and the exocrine-like subtypes [7]. Using a different gene expression microarray, Moffit et al. distinguished the classical and basal-like subtypes [8]. Puleo et al. identified the undifferentiated subtype; the pure basal-like subtype; two moderately differentiated subtypes, the stroma-activated subtype and the desmoplastic subtype; and a well-differentiated subtype, the pure classical variant [9].

More recently, Chan-Seng-Yue et al. separated the classical and basal-like subtypes into two categories, such as A and B, and included a fifth subtype, the hybrid subtype. Moreover, they correlated transcriptomic expression to the stage of disease at diagnosis [10]. Despite some differences in these classifications, there were significant overlaps between the classical/pancreatic progenitor and quasi-mesenchymal/basal-like/squamous subtypes signatures described by Moffit, Collinson, Bailey, Puleo, and Chan-Seng-Yue [11]. The information acquired by these studies may help to distinguish PDAC in at least two different variants, with different biological behavior, metabolism, proteomic expression, disease biomarkers, and therapeutic approaches. Moreover, the implementation of this evidence in clinical trials may help to direct the efforts of drug development toward specific biological features of the disease.

However, the complexity, high expertise required, and costs of the omics approach are the main limitations for its use in clinical practice at present.

The present review summarizes the main evidence of the application of omics sciences to the study of PC and the identification of future perspectives.

## 2. Metabolic Changes in Pancreatic Cancer Cells

In response to intrinsic oncogenic alterations and the influence of the tumor microenvironment, cancer cells go through a complex oncogene-directed metabolic reprogramming that includes changes in glucose utilization, lipid and amino acid metabolism, redox balance, and activation of recycling and scavenging pathways [12].

Reprogramming of energy metabolism was recently recognized as an emerging hallmark of cancer [13]. In particular, cancer cells shift their glucose metabolism to glycolysis, even in the presence of a sufficient oxygen supply (“aerobic glycolysis”) in order to sustain various biosynthetic pathways through the production of different intermediate glycolysis products [13]. Besides the well-characterized reprogramming of glucose and glutamine metabolism, which is characterized by the up-regulation of glycolysis, non-oxidative pentose phosphate pathway, hexosamine biosynthesis pathway, and glutamine anaplerosis [14,15], lipid metabolism has been increasingly recognized as a crucial factor in cancer pathogenesis. In cancer, lipid metabolism changes are involved in both anabolic and catabolic pathways in order to sustain a high rate of membrane biosynthesis during proliferation, energy demands, and altered signaling pathways [16].

Cancer cells activate lipogenesis, which is physiologically restricted to hepatocytes and adipocytes. Cancer cells increase acetyl-Coenzyme-A (acetyl-CoA) production via adenosine triphosphate (ATP)-citrate lyase in the tricarboxylic acid cycle or via acetyl-CoA synthetase to synthesize de novo fatty acids and cholesterol. Similarly, acetyl-CoA carboxylase, the rate-limiting enzyme for fatty acid synthesis, fatty acid synthase (FAS), which produces 16-carbon palmitate for the synthesis of more complex fatty acids and membrane components, and stearoyl-CoA desaturase, which produces monounsaturated fatty acids and palmitoleic acid, were associated with pancreatic tumorigenesis and progression [17]. In contrast to lipogenesis, PC cells also promote the lipolytic pathway to generate ATP and other molecules involved in biosynthetic reactions, overexpress enzymes involved in fatty acid oxidation [18], and reduce lipid peroxidation to inhibit death mechanisms, such as ferroptosis [19]. Many of these alterations were associated with the Kirsten rat sarcoma (KRAS) virus, the most prevalent oncogene contributing to pancreatic tumorigenesis, suggesting that metabolic rearrangement may be a consequence of oncogene-induced modifications [19,20,21]. However, conflicting results regarding the up-regulation of certain metabolic pathways, such as lipid uptake or the variable abundance of specific molecules in different PC cell lines [22], support the hypothesis that different cancer subtypes have significant genetic and metabolic heterogeneity [23].

## 3. Metabolomics

In response to intrinsic oncogenic alterations and the influences of the tumor microenvironment, cancer cells go through a complex oncogene-directed metabolic reprogramming that includes changes in glucose utilization, lipid and aminoacidic metabolism, redox balance, and activation of recycling and scavenging pathways [12] (Table 1). Consequently, metabolic subtypes of PC have been identified. Based on extended metabolic profiling in cell line models, Daemen et al. described three PC subtypes: the low proliferating, glycolytic, and lipogenic subtypes. The low proliferating subtype exhibited low amino acid and carbohydrate levels. The glycolytic subtype revealed an enrichment of glycolysis and serine pathway components. The lipogenic subtype showed a high abundance of different lipid metabolites. Interestingly, the glycolytic and lipogenic subtypes were associated with the previously identified molecular quasimesenchymal/basal-like and classical subtypes, respectively [24,25]. Moreover, a correlation between these metabolic subtypes and other molecular subtypes was also observed [7], strengthening the relation between molecular and metabolic subtypes. Using 77 patient-derived tumor xenografts, Kaoutari and colleagues discovered a metabolic signature exhibiting a significant correlation with transcriptomic phenotypes [26]. In another study that analyzed the immunohistochemical expression of metabolism-related proteins, Yu et al. identified eight subtypes of PC that were classified into two categories: glucose-dependent and glutamine-dependent. Among these eight subtypes, the Warburg type, non-canonical type, and mixed type were metabolically active, biologically aggressive, and associated with a poorer prognosis [27]. These findings suggest that distinct metabolic rearrangements occur during carcinogenesis and that these modifications are associated with specific biological behaviors.

Considering the metabolic phenotypes of PC that have been recently identified, products resulting from the impaired metabolism of cancer cells may be detected in body fluids and used as potential biomarkers of the disease. Recent advances in the feasibility and reproducibility of metabolomic techniques have enhanced its use in the research field of biomarker identification for PC diagnosis and prognosis.

Different metabolomic-based panels showed very high accuracy in discriminating PC from healthy controls. Analyzing the saliva of 215 subjects, of which 18 had PC, 69 had oral cancer, 30 had breast cancer, 11 had periodontal disease, and 87 were healthy controls, Sugimoto et al. identified 48 metabolites as candidate biomarkers for PC with an area under the curve (AUC) of 0.993 to discriminate PC patient from healthy individuals [28]. In a meta-analysis, Mehta and colleagues chose 10 common blood metabolites associated with PC that were reported in independent studies. The 10-biomarker panel showed an AUC of 0.997, 0.992, and 0.653 to discriminate PC from normal controls, diabetic patients, and colorectal cancer patients, respectively [29]. These results suggest that measuring the metabolic changes in cancer may be a useful tool to distinguish cancerous and non-cancerous conditions, although there may be overlaps across different types of cancer.

Along the same line, Luo et al. identified plasma biomarkers that precisely distinguished PC patients from healthy controls using a precision-targeted metabolomics method. This panel included creatine, inosine, beta-sitosterol, sphinganine, and glycocholic acid. Moreover, two plasma biomarkers, succinic acid and gluconic acid, were identified to efficiently diagnose the progression and metastasis of PC, showing the potential of metabolomics not only for diagnosis but also for the detection of disease progression [30].

The specificity of metabolomic signatures is crucial for accurately distinguishing between patients with PC and patients with other pancreatic diseases, particularly those conditions that represent a risk factor for PC. In a large multicentre prospective metabolome investigation including 271 patients with PC and 282 with chronic pancreatitis (CP), Mayerle et al. identified a composite panel of biomarkers (MxPancreasScore) that was able to distinguish all stages of PC and resectable PC from CP with a very high accuracy (90.6% and 90.8%, respectively). In both cases, the performance of this biomarker signature in detecting PC was significantly better compared to that of the carbohydrate antigen (CA 19-9) (AUC 0.94 vs. 0.85 (*p* < 0.001) for all tumor stages; 0.93 vs. 0.84 (*p* < 0.001) for resectable PC). In addition to CA 19-9, this panel included nine metabolites, five of which belonged to the ontology class of lipids and included sphinganine-1-phosphate, two sphingomyelins, and one ceramide [31]. More recently, the improved metabolic signature (i-Metabolic) and minimalistic metabolic signature (m-Metabolic) were developed and validated to overcome the limitations associated with methods and costs using single-platform and single-run techniques. The i-Metabolic comprised CA 19-9 and twelve metabolites, including four sphingomyelins and two ceramides, while m-Metabolic comprised CA 19-9 and four metabolites, including one sphingomyelin and one ceramide [32]. These results underscore the significance of lipids and their subclasses as one of the most promising classes of metabolites in the field of metabolomic biomarker research.

Supporting this, a metabolome and lipidome analysis on 361 patients with PC from three different cohorts revealed 497 common metabolites, 76% of which belonged to the class of complex lipids and fatty acids, with sphingomyelins, ceramides, triacylglycerol, and phosphatidylcholines being the most abundant lipid classes. Analysis of the abundance and composition of these complex lipids led to the identification of three distinct metabolic subtypes. Specifically, subtype 1 was characterized by a decrease in ceramides and a strong enrichment in triacylglycerol, subtype 2 was characterized by a high abundance in ceramides and triacylglycerol and a reduction in sphingomyelins, and subtype 3 was characterized by a significant decrease in ceramides and triacylglycerols and a variating spectrum of sphingomyelin species. The analysis of metabolic pathways revealed that sphingolipid pathways differed among the three subtypes, indicating that sphingolipids may play a crucial role in delineating metabolic PC subtypes [33].

The identification and validation of innovative and reliable biomarkers are essential for progressing towards personalized medicine. Nevertheless, there is a substantial gap between the findings of cancer biomarker studies and their translation into potential clinical applications [34]. The elucidation of biochemical alterations underpinning cancer progression, as reflected by biomarkers, allows a pragmatic pathway for delineating molecular targets conducive to a tailored therapeutic approach.

**Table 1 ijms-25-07623-t001:** Main studies on metabolomics in pancreatic cancer.

Article	Results
Sugimoto et al. (2010) [28]	Analysis of 57 different metabolites in saliva samples using capillary electrophoresis time-of-flight mass spectrometry to discriminate PDAC patients from healthy controls with an AUC of 0.993.
Daemen et al. (2015) [24]	Identification of three metabolic subtypes in PDAC: low proliferating (low amino acid and carbohydrate levels), glycolytic (enrichment of glycolysis and serine pathway components, association with mesenchymal subtype), and lipogenic (abundance of different lipid metabolites, association with epithelial subtype).
Yu et al. (2015) [27]	Glucose-dependent metabolic (Warburg and mixed) subtypes associated with nerve infiltration (*p* = 0.0003), UICC stage (*p* = 0.0004), activated autophagic status in tumor (*p* = 0.0167), positive marginal status (*p* < 0.0001), lymphatic invasion (*p* < 0.0001), and activated autophagic status in stroma (*p* = 0.0002), respectively. Glutamine-dependent metabolic (non-canonical and mixed) subtypes associated with vascular invasion (*p* = 0.0073), highest percentage of activated autophagy in tumors (*p* = 0.0034), and shorter overall survival (*p* < 0.001) in PDAC.
Mehta et al. (2017) [29]	Identification of a panel of 10 blood metabolites using targeted mass spectrometry to discriminate PDAC from healthy control (AUC = 0.997) patients with type 2 diabetes mellitus (AUC = 0.992) and colorectal cancer patients (AUC = 0.653).
Mayerle et al. (2018) [31]	Identification of a composite panel of biomarkers (9 metabolites—class of lipids including sphinganine-1-phosphate, two sphingomyelins, and one ceramide) to distinguish all stages of PDAC and resectable PDAC from CP with higher accuracy (90.6% and 90.8% respectively) than CA 19-9 (AUC 0.94 vs. 0.85, *p* < 0.001 for all tumor stages; 0.93 vs. 0.84, *p* < 0.001 for resectable PDAC).
Luo et al. (2020) [30]	Analysis of five metabolite biomarkers in plasma (creatine, inosine, beta-sitosterol, sphinganine and glycocholic acid) with higher accuracy and specificity to diagnose PDAC than conventional biomarkers (CA 125, CA 19-9, CA 242, and CEA). Role of succinic acid and gluconic acid in monitoring progression and metastasis of PDAC at different stages.
Kaoutari et al. (2021) [26]	Association of a metabolic signature with PDAC molecular gradient (R = 0.44 and *p* < 0.001) to predict clinical outcomes (*p* < 0.001, HR = 2.68, 95% CI: 1.5–4.9), transcriptomic phenotypes, and drug resistance (gemcitabine, oxaliplatin, docetaxel, SN-38, and 5-Fluorouracil).
Mahajan et al. (2021) [33]	Identification of three metabolic PDAC subtypes associated with distinct complex lipid patterns: subtype 1 (reduced ceramide levels, strong enrichment of triacylglycerols), 2 (increased abundance of ceramides, sphingomyelin, and other complex sphingolipids), and 3 (decreased levels of sphingolipid metabolites).
Mahajan et al. (2022) [32]	Role of i-Metabolic Signature (12 analytes + CA 19-9) in distinguishing PDAC from CP with AUC of 97.2%, 93.5%, and 92.2% in the identification and validation of cohorts 1 and 2, respectively. Role of m-Metabolic signature (4 analytes + CA 19-9) in discriminating PDAC from CP with a sensitivity of 77.3%, a specificity of 89.6%, and an overall accuracy of 82.4%.

Abbreviations: AUC, area under the curve; CI, confidence interval; CP, chronic pancreatitis; HR, hazard ratio; PDAC, pancreatic ductal adenocarcinoma; UICC, union for international cancer control.

## 4. Proteomics

Proteomics is a rapidly growing field of biomedical research. The studies of proteome-based biomarkers seek to identify molecular profiles in tumors that may have a diagnostic, predictive, and/or prognostic value. Current interest in proteomics is due to the opportunities it can offer in terms of delineating altered protein expression, developing new biomarkers for the early diagnosis of diseases, and identifying new therapeutic targets. The main approaches include comparative analysis of protein expression in normal and pathological tissues to identify aberrantly expressed proteins that may represent novel markers, analysis of secreted proteins (secretome) in cell lines and primary cultures, and direct analysis of the serum protein profile [35,36] (Table 2).

Papapanagiotou et al. recently investigated the role of secreted protein acidic and rich in cysteine (SPARC) and osteonectin in various processes during the development of pancreatic adenocarcinoma, showing a sensitivity of 84.6% and a specificity of 87.5% for the detection of early-stage PDAC [37]. Similarly, ZIP4 is a significantly up-regulated zinc membrane transporter in neoplastic PC cell lines with an important role in promoting tumor growth and proliferation. It was also isolated on the membranes of circulating exosomes [38]. On the contrary, β-catenin is a protein expressed both at the cytoplasmic and at the membrane level that seems to be associated with a lower risk of disease progression, as shown in a recent study [39].

Many studies evaluated the potential diagnostic role of proteins as PDAC biomarkers alone or in combination with CA 19-9. A study reported significantly higher levels of alcohol dehydrogenase (ADH) and circulating macrophage inhibitory cytokine (MIC-1) in PDAC patients than in healthy controls, with a sensitivity of 62% and 94%, respectively, and a specificity of 87.5% and 45%, respectively. The study also showed improved efficacy of diagnosis when combined with CA 19-9, with an AUC of 0.89 [40].

The assessment of protein panels in serum or plasma using ELISA or proximity ligation assay (PLA) has been applied in different studies to evaluate biomarkers alone or in composite diagnostic panels, with or without CA 19-9, to reach superior sensitivity compared to CA 19-9 alone in distinguishing patients with PDAC, benign pancreatic conditions, and healthy controls.The combination of tissue inhibitors, metalloproteinase-1 (TIPM1), leucine-rich alpha-2 glycoprotein 1 (LRG1), and CA 19-9, reached a greater sensitivity (84.9%) and specificity (95%) in discriminating early-stage PDAC than CA 19-9 alone [41].

Moreover, insulin-like growth factor-binding protein 2 (IGFBP2) and insulin-like growth factor-binding protein 3 (IGFBP3) were reported to serve as compensatory biomarkers for CA 19.9, showing a sensitivity of 68.4% and 76.3%, and a specificity of 67.7% and 70.7%, respectively, when used alone for detecting early-stage PDAC, and an increased effectiveness of detection (AUC 0.90) when used in combination with CA 19-9 (AUC 0.90) [42].

A composite panel of CA 19-9, OPN, and CHI3L1 was demonstrated to have superior sensitivity for PC compared with CA 19-9 alone (93% vs. 80%), whereas CEA and CA 125 showed prognostic significance for the survival of PC (*p* < 0.003) when measured simultaneously [43].

A panel composed of CA 19-9, CEA, and TIMP-1 was tested by Brand et al. to discriminate PDAC patients from benign subjects, with a high sensitivity and specificity both in the training and in the validation set. Similarly, they tested a panel with CA 19-9, ICAM-1, and OPG, which demonstrated selectivity for PDAC and did not recognize breast (specificity = 100%), lung (specificity = 97%), or colon (specificity = 97%) cancer [44].

Among many proteins demonstrated to differ in terms of expression between non-tumor and tumor samples, McKinney et al. analyzed biglycan (BGN), pigment epithelium-derived factor (PEDF), thrombospondin-2 (THBS-2), and TGF-β induced protein ig-h3 precursor (βIGH3) as potential minimally invasive diagnostic markers for their effect in cancer when up-regulated [45].

In the differentiation of pancreatic neoplasia non-neoplastic pancreatic disease, based on SELDI-TOF mass spectrometry, Ehmann et al. proposed a model based on apolipoprotein A-II, transthyretin and apolipoprotein A-I that resulted in decreased levels in cancer sera. The model showed a sensitivity of 100% and a specificity of 98% for the training data, and a sensitivity of 83% and a specificity of 77% for test data [46].

The comparison of protein expression between neoplastic and non-neoplastic tissues may also help to identify proteins specifically expressed in the tumor that may be possible targets for future drug development. In a recent study, mesothelin, mucin 4 (MUC4), annexin A10 (ANXA10), and glypican 1 (GPC-1) were observed to be selectively expressed in the tumor when compared to non-neoplastic pancreatic ducts and acini [47].

The identification of highly expressed proteins that are associated with prognosis is another field of application of proteomics. Indeed, the prediction of rapid progression and response to treatments is an important issue in the allocation of therapies in the era of precision medicine.

Fibrillar collagen COL6A3, fibrillin-1 (FBN-1), fibronectin (FN1), fibrinogens (FGA, FGB, FGG), periostin (POSTN) and members of the S100 family, are over-expressed in pancreatic neoplastic tissue, whereas others belonging to the family of small leucine-rich proteoglycans (SLRPs) such as osteoglycin (OGN), PRELP, fibromodulin (FMOD), decorin (DCN), and asporin (ASPN) are associated with a worse prognosis [48].

A recent membrane proteomics and tissue microarray study evaluated the role of brain acid soluble protein 1 (BASP1), a protein implicated in the biochemical pathway of Wilms tumor protein (WT1), in predicting the outcome of patients with PC, showing its association with prolonged survival and a better response to adjuvant chemotherapy treatment than in patients in whom the same marker is negative [49].

The overexpression of the TROP2 (human trophoblast cell-surface) antigen and the low expression of the junctional adhesion molecule A (JAM-A) antigen were found to be significantly associated with decreased OS and correlated with lymph node metastasis and tumor grade [50].

Zong et al. investigated the expression of T-box transcription factor 4 (TBX4) in stage II PDAC tumors, showing a significant correlation between TBX4 expression and tumor grade, liver metastasis recurrence, and longer patient survival [51].

Tissue microarray analysis seems to be a useful tool to assess the expression of heat shock protein 27 (HSP27) that has been demonstrated to have an important impact on overall survival (OS) and chemo- and radio-sensitivity of PDAC cells. Higher levels of HSP27 make cells more susceptible to treatment with gemcitabine, while lower levels of HSP27 are associated with gemcitabine resistance [52].

A recombinant enzyme, deoxycytidine kinase (dCK), was identified by Maréchal et al. to be associated with prolonged survival after adjuvant gemcitabine administration for resected PDAC [53].

Components of the Notch pathway (Notch1, Notch3, Notch4, HES-1, HEY-1) were examined both in resectable and inoperable pancreatic cancers. They were significantly increased in tumor tissue and in locally advanced and metastatic tumors compared to resectable cancers. Moreover, nuclear Notch3 and HEY-1 expression was shown to be significantly associated with reduced OS and DFS following tumor resection [54].

Proteomic profiling has also been used to study the tumor microenvironment. In particular, the analysis of immune cells and cytokine expression has been a main focus, since it may provide the basis for the application of immunotherapy. In this context, CD9 and CD63 were used as potential markers for the distinction of inflammatory, precancerous, and malignant lesions at the pancreatic level, as they are involved in the transformation of healthy cells into neoplastic ones; however, the studies produced so far on these antigens are burdened by a low sample size [55]. Preoperative levels of IL-6 and CA 19-9 were independently associated with shorter OS in inoperable patients [56].

Biological analysis and cytological validation by Liu et al. confirmed in their in vitro study that knockdown of circulating chaperonin-containing T-complex protein 8 (CCT8) may suppress the metastatic phenotype of PDAC cell lines [57]. The predictive nature of insulin-like growth factor (IGF) on the treatment effect of chemotherapy (Ganitumab + Gemcitabine) in metastatic PDAC was assessed by McCaggery et al., who found improved OS after the treatment in patients with higher levels of IGF-1, IGF-2, or IGFBP-3, or lower levels of IGFBP-2 [58].

**Table 2 ijms-25-07623-t002:** Main studies on proteomics in pancreatic cancer.

Article	Proteins	Results
Papapanagiotou et al. (2018) [37]	SPARC, Osteonectin	Sensitivity of 84.6% and specificity of 87.5% in detection of early-stage PDAC.
Jin et al. (2018) [38]	ZIP4	Discrimination between malignant pancreatic cancer patients and benign pancreatic disease patients with an AUC of 0.89.
Saukkonen et al. (2016) [39]	PROX1, β-catenin	High PROX1 (48%) and β-catenin (65%) expression in PDAC associated with lower risk of death from PDAC (HR = 0.63; 95% CI, 0.42–0.95, *p* = 0.026; and HR = 0.54; 95% CI, 0.35–0.82, *p* = 0.004; respectively). Combined high expression predicting lower risk of death from PDAC (HR = 0.46; 95% CI, 0.28–0.76, *p* = 0.002).
Mohamed et al. (2016) [40]	ADH, MIC-1	High sensitivity (90%) and specificity (83%) for ADH in detecting early PDAC. Improved efficacy when ADH and MIC-1 combined to CA 19-9 (*p* = 0.023, AUC 0.89).
Capello et al. (2017) [41]	TIPM1, LRG1, CA 19-9	Improvement of sensitivity (0.849 vs. 0.667) at 95% specificity with an AUC of 0.949 (95% CI, 0.92–0.98) and 0.887 (95% CI, 0.82–0.96) in discriminating early-stage PDAC vs. healthy subjects in combined validation and test sets, respectively. Better performance compared to CA 19-9 alone (*p* < 0.001 combined validation set; *p* = 0.008 test set).
Yoneyama et al. (2016) [42]	IGFBP2, IGFBP3	Sensitivity of 68.4% and 76.3% and specificity of 67.7% and 70.7%, respectively, for IGFBP2 and IGFBP3 in detecting early-stage PDAC. IGFBP2 associated with increased risk of diseases of pancreatic malignancy. Combination of IGFBP2 and IGFBP3 with CA 19-9 with an AUC of 0.90.
Chang et al. (2009) [43]	Osteopontin, Chitinase 3-like 1, CA 19-9	Higher sensitivity for PDAC compared with CA 19-9 alone (93% vs. 80%). CEA and CA 125 with prognostic significance for survival for local advanced PDAC (*p* < 0.003).
Brand et al. (2011) [44]	CA 19-9, CEA, TIMP-1	Higher sensitivity (respectively 76% and 71%) and specificity (respectively 90% and 89%) in discriminating PDAC from benign subjects in training tests and independent validation sets.
McKinney et al. (2011) [45]	BGN, PEDF, THBS-2, βIGH3	Up-regulation in BGN, PEDF, THBS-2, and βIGH3 associated with PDAC progression, as players in tumor microenvironment, cell proliferation, or angiogenic processes.
Ehmann et al. (2007) [46]	Apolipoprotein A-II, transthyretin, apolipoprotein A-I	Sensitivity of 100% and specificity of 98% for training data set and sensitivity of 83% and specificity of 77% for test data in differentiation of PDAC from healthy controls.
Nicoletti et al. (2023) [47]	MSLN, MUC4, ANXA10, GPC-1	Selective expression of MSLN, MUC4, ANXA10, and GPC-1 in the neoplastic tissue compared to non-tumor ductal and acinar tissues (*p* < 0.001).
Tian et al. (2019) [48]	Fibrillar collagen COL6A3, FBN-1, FN1, fibrinogens, POSTN, PRELP, FMOD, DCN, OGN, ASPN	Overexpression of COL6A3, FBN-1, FN1, fibrinogens (FGA, FGB, and FGG), and POSTN in PDAC. OGN, PRELP, FMOD, DCN, and ASPN associated with worse prognosis.
Zhou et al. (2019) [49]	BASP1, WT1	BASP1 association with prolongation of survival (HR 0.468, 95% CI, 0.257–0.852, *p* = 0.013) and better response to adjuvant chemotherapy treatment in PDAC. WTI association with worsened survival (HR 1.636, 95% CI, 1.083–2.473, *p* = 0.019) and resistance to chemotherapy.
Fong et al. (2008) [50]	TROP2, JAM-A	TROP2 overexpression associated with decreased overall survival (*p*< 0.01), presence of lymph node metastasis (*p* = 0.04), tumor grade (*p* = 0.01), and poor progression-free survival after surgery (*p* < 0.01).
Zong et al. (2011) [51]	TBX4	High expression (62.3%) in PDAC associated with longer survival (*p* = 0.010).
Schafer et al. (2012) [52]	HSP27	High expression in PDAC correlated inversely with nuclear p53 accumulation and associated with better response to chemotherapy with Gemcitabine.
Marechal et al. (2010) [53]	dCK	Association with prolonged survival after adjuvant Gemcitabine for resected pancreatic adenocarcinoma as independent prognostic factor (DFS: HR, 3.48; 95% CI, 1.66–7.31; *p* = 0.001; OS: HR, 3.2; 95% CI, 1.44–7.13; *p* = 0.004).
Mann et al. (2012) [54]	Notch1, Notch3, Notch4, HES-1, HEY-1	Increased expression in tumor tissue and locally advanced and metastatic PDAC compared to resectable PDAC (*p* ≤ 0.001). Notch3 and HEY-1 expression associated with reduced OS and DFS following tumor resection.
Khushman et al. (2017) [55]	CD63, CD9	Higher expression in pathologic tissues compared with adjacent normal tissues (mean multiplicative Q score with *p* = 0.0041 and *p* = 0.0018; mean Q score with *p* < 0.0001 and *p* < 0.0124).
Schultz et al. (2015) [56]	YKL-40, IL-6, CA 19-9	Significant OR for prediction of PDAC: - CA 19-9: OR = 2.28, 95% CI, 1.97–2.68, *p* < 0.0001, AUC = 0.94. - YKL-40: OR = 4.50, 95% CI, 3.99–5.08, *p* < 0.0001, AUC = 0.87. - IL-6: OR = 3.68, 95% CI, 3.08–4.44, *p* < 0.0001, AUC = 0.87. Association with short overall survival of high preoperative: - CA 19-9: HR = 2.51, 95% CI, 1.22–5.15, *p* = 0.013. - IL-6: HR = 2.03, 95% CI, 1.11–3.70, *p* = 0.021. and pre-treatment levels - YKL-40: HR = 1.30, 95% CI, 1.03 to 1.64, *p* = 0.029. - IL-6: HR = 1.71, 95% CI, 1.33–2.20, *p* < 0.0001. - CA 19-9: HR = 1.54, 95% CI, 1.06–2.24, *p* = 0.022.
McCaffery et al. (2013) [58]	IGF-1, IGFBP2-3	Improved OS association in treatment with Ganitumab with higher levels of IGF-1 (16 vs. 6.8 months-HR, 0.25; 95% CI: 0.09–0.67), IGF-2 (16 vs. 5.9 months-HR, 0.24; 95% CI: 0.09–0.68), and IGFBP-3 (16 vs. 6.8 months-HR, 0.28; 95% CI: 0.11–0.73), or lower levels of IGFBP-2 (12.7 vs. 6.6 months-HR, 0.19; 95% CI: 0.07–0.55) in PDAC.

Abbreviations: βIGH3, TGF-β induced protein ig-h3 precursor; ADH, alcohol dehydrogenase; ANXA10, annexin A10; ASPN, asporin; AUC, area under the curve; BASP1, brain acid soluble protein 1; BGN, biglycan; dCK, deoxycytidine kinase; DCN, decorin; DFS, disease-free survival; FBN-1, fibrillin-1; FMOD, fibromodulin; FN1, fibronectin; GPC-1, glypican 1; HR, hazard ratio; HSP27, heat shock protein 27; IGF, insulin-like growth factor; IGFBP, insulin-like growth factor-binding protein; JAM-A, junctional adhesion molecule A; LRG1, leucine-rich alpha-2 glycoprotein 1; MIC-1, circulating macrophage inhibitory cytokine; MSLN, mesothelin; MUC4, mucin 4; OGN, osteoglycin; OR, odds ratio; PEDF, pigment epithelium factor; PDAC, pancreatic ductal adenocarcinoma; POSTN, periostin; TBX4, T-box transcription factor 4; SPARC, secreted protein acidic and rich in cystein; WT1, Wilms tumor protein.

## 5. Genomics

In the last few years, many studies focused on the genomic landscape of PC and its potential use in clinical practice (Table 3).

Mutations in *KRAS* (*KRAS-MUT*) are found in 85–90% of PCs and are one of the major early driver events in the disease development. Afterwards, other mutations occur, including the loss of TP53 and inactivating mutations in *CDKN2A* and/or *SMAD4* [59,60,61].

The remaining 10–15% of *KRAS* wild-type (*KRAS-WT*) tumors harbor frequent genomic events that can be targetable, such as homologous recombination genes, including *BRCA1/2*, *PALB2*, mismatch repair deficiency (dMMR), *BRAF* mutations/fusions, *NRG1* and *NTRK* fusions, and other less common mutations [62].

Moreover, a direct correlation between *KRAS-WT PC* and an early-onset type of disease, defined as an occurrence within 50 years of age, was observed in these studies [63,64].

*C-Myc* overexpression is another common finding in PC and predicts the aggressive behavior of cancer cells. It binds the promoter of different genes, thereby regulating their transcription. *C-Myc* overexpression is associated with chemoresistance, intra-tumor angiogenesis, epithelial–mesenchymal transition (EMT), and metastasis in PC [65].

More recently, *NCAPG2* has emerged as an intrinsically essential participant of the condensin II complex, which is involved in the process of chromosome cohesion and stabilization in mitosis. *NCAPG2* was identified to be overexpressed in almost every tumor and exhibited significant prognostic and diagnostic efficacy. Furthermore, the correlation between *NCAPG2* and selected immune features, such as immune cell infiltration, immune checkpoint genes, tumor mutational burden (TMB), microsatellite instability (MSI), and other oncological features, indicates that *NCAPG2* could potentially be applied in the guidance of immunotherapy. Down-regulation of *NCAPG2* expression is associated with reduced proliferation, invasion, and metastasis of PC cells [66].

In a personalized approach, genomic analysis can have direct applications in treatment in clinical practice. Aside from standard chemotherapy in the advanced stages of PC, such as FOLFIRINOX and gemcitabine/nab-paclitaxel, personalized approaches have been explored in disease treatment since the approval of the PARP inhibitor, olaparib for germline *BRCA1/2* mutations in 2019 [67].

The *Know Your Tumour Initiative* demonstrated a survival benefit in patients with PDAC receiving a genomic-matched approach. Patients with actionable molecular alterations who received matched therapy (n = 46) had significantly longer median OS than those patients who only received unmatched therapies because they did not have an actionable molecular alteration (2.58 years vs. 1.51 years) [68].

Several studies showed promising results for the inhibition of *KRAS-MUT*. *MRTX1133*, a potent, selective, and non-covalent *KRAS^G12D^* inhibitor, exhibited a dose-dependent inhibition of *KRAS*-mediated signal transduction and marked tumor regression (≥30%) in a subset of *KRAS^G12D^*-mutant cell line-derived and patient-derived xenograft models, including 8/11 (73%) PDAC models [69].

In another study by Strickler et al., the safety and efficacy of sotorasib, a *KRAS^G12C^* inhibitor, was analyzed in 38 patients with *KRAS p.G12C*-mutated PC who had received at least one previous systemic therapy [70].

Finally, adagrasib, another *KRAS^G12C^* inhibitor already in use in *KRASG12C*-mutated non-small-cell lung cancer (NSCLC) and colorectal cancer (CRC), was tested with partial response in pancreatic and biliary tract cancers [71].

In a targeted multi-gene assay analysis [72], among 795 exocrine PC, 73 patients had *KRAS-WT* tumors (9.2%), 43.8% (32/73) of which had *MAPK* pathway alterations. There were 18 *BRAF* alterations, most of which were known to be activating, including a *p.N486_P490*del in-frame deletion (n = 9), *BRAFV600E* mutation (n = 3), and BRAF fusion (n = 1). In addition, there were seven receptor tyrosine kinase (RTK) fusions, involving *ROS1* (n = 2), *NRG1* (n = 2), *NTRK1* (n = 1), *NTRK3* (n = 1), *FGFR2* (n = 1), and seven other *MAPK* pathway alterations, including amplifications in *EGFR* (n = 1), *ERBB2* (n = 1), and *MET* (n = 1). The authors also demonstrated proof-of-principle for sensitivity to *MAPK* pathway targeting in *KRAS WT* subsets. Organoid models derived from two patients with BRAF in-frame deletions responded to dual pan-RAF inhibitor (LY3009120) and MEK inhibitor (trametinib) treatment.

A patient harboring a *ROS1 fusion* (SLC4A4::ROS1) demonstrated sustained clinical benefit with crizotinib and cabozantinib, targeted therapies with activity against ROS1, providing clinical evidence for the feasibility of this strategy.

Among the remaining 56.2% (41/73) of *KRAS-WT* tumors without a *MAPK* pathway alteration, 29.3% (12/41) harbored activating oncogenic alterations in other drivers including *GNAS* (n = 6), *MYC* (n = 2), *PIK3CA*, and *CTNNB1*. Preferential loss of the tumor suppressor *PTEN* was also observed in *KRAS-WT* cases [18].

Besides the study of oncogenes, the interest in the study of non-coding genetic material, particularly micro-RNA, has recently increased [73]. Since circulating genetic material is rapidly cleared, extracellular vesicle (EV)-derived micro-RNAs are a more reliable biomarker of disease [74]. A panel of 11 micro-RNA (miR-21, -34a, -99a, -100, -125b, -148a, -155, -200a, -200b, -200c, and -1246) was tested as a diagnostic and prognostic tool for PC. The expression of miR-200b and miR-200c was associated with PC compared to chronic pancreatitis (*p* = 0.005; *p* = 0.19) and controls (*p* < 0.001; *p* = 0.024). When combined with levels of CA 19-9, a diagnostic accuracy of 97% was reached. Moreover, the expression of miR-200c correlated with a poorer OS [75]. The association of miR-200b with PC was also confirmed in another study [76]. Exosomal miR-21, miR-10b, and miR-17-5p were also associated with PC [77,78].

**Table 3 ijms-25-07623-t003:** Main studies on genomics in pancreatic cancer.

Article	Imaging	Results
Hosein et al. (2022) [59]	KRAS, P53, CDKN2A, SMAD4, BRCA1/2, PALB2, dMMR, BRAF, NRG1, NTRK	KRAS, TP53, CDKN2A, and SMAD4 mutations present in >90% of patients with PDAC. Association of chromatin modification genes (ARID1A, KMT2D, and KMT2C), DNA repair genes (BRCA1, BRCA2, and PALB2), and additional oncogenes (BRAF, MYC, FGFR1) in about 10% of patients with PDAC.
Varghese et al. (2021) [63]	ETV6-NTRK3, TPR-NTRK1, SCLA5-NRG1, ATP1B1-NRG1 fusions, IDH1 R132C mutation, mismatch repair deficiency	Association of KRAS wild-type cancers (ETV6-NTRK3, TPR-NTRK1, SCLA5-NRG1, and ATP1B1-NRG1 fusions, IDH1 R132C mutation, and mismatch repair deficiency) with early-onset of disease.
Ben Aharon et al. (2019) [64]	SMAD4	Association of SMAD4 higher mutation rates, higher expression levels of phospo-GSK3 and increased activation of TGFb pathway with early-onset PDAC.
Ala et al. (2021) [65]	C-Myc	Association of C-Myc overexpression with chemoresistance, intra-tumor angiogenesis, epithelial–mesenchymal transition, metastasis, and aggressive behavior of PDAC.
Wang et al. (2023) [66]	NCAPG2	Association of NCAPG2 overexpression with immune cell infiltration, immune checkpoint genes, tumor mutational burden, and microsatellite instability. Association of NCAPG2 down-regulation with reduced proliferation, invasion, and metastasis in PDAC.
Golan et al. (2019) [67]	BRCA1, BRCA2	Association of Olaparib (PARP inhibitor) treatment with longer median progression-free survival than in the placebo group (7.4 months vs. 3.8 months; HR for disease progression or death, 0.53; 95% confidence interval, 0.35 to 0.82; *p* = 0.004).
Hallin et al. (2022) [69]	KRAS	Association of KRAS mutant inhibitors (MRTX1133) with marked tumor regression (≥30%) in PDAC.
Strickler et al. (2023) [70]	KRAS	Association of KRAS G12C inhibitor (Sotorasib) with anticancer activity and acceptable safety profile in advanced PDAC that had received previous treatment.
Bekaii-Saab et al. (2023) [71]	KRAS	Association of KRAS G12C inhibitor (Adagrasib) with encouraging response (median progression-free survival of 7.4 months—95% CI, 5.3 to 8.6) and good tolerance in pretreated PDAC patients.
Garcia et al. (2017) [72]	BRAF, RTK, MAPK	Association of BRAF alterations (p.N486_P490del in-frame deletion, BRAFV600E mutation, BRAF fusion), receptor tyrosine kinase (RTK) fusions (ROS1, NRG1, NTRKQ, NTRK3, and FGFR2), and MAPK pathway alterations (amplifications in EGFR, ERBB2, and MET) with PDAC.
Reese et al. (2020) [75]	miR-200b, miR-200c	Association of overexpression of miR-200b and miR-200c with PDAC as compared to healthy controls (*p* < 0.001; *p* = 0.024) and CP (*p* = 0.005; *p* = 0.19). Correlation of high expression of miR-200c and miR-200b with shorter overall survival (*p* = 0.038 and *p* = 0.013 respectively).
Li et al. (2010) [76]	miR-200a, miR-200b	Association of miR-200a and miR-200b hypomethylation and overexpression with PDAC. Association of elevated levels of miR-200a and miR-200b with PDAC and CP compared with healthy controls (*p* < 0.00019).
Pu et al. (2020) [77]	miR-21, miR-212-3p, miR-10b	Association of higher levels of miR-21 and miR-10b with PDAC. miR-21 with better diagnostic performance (*p* = 0.0003, AUC 0.72). Better diagnostic value with combination of miR-21 and miR-10b (*p* < 0.0001, AUC 0.79). Role of miR-21 in distinguishing early-stage PDAC from control and advanced-stage PDAC (*p* < 0.05, early-stage vs. healthy; *p* < 0.001, early-stage vs. advanced stage).
Que et al. (2013) [78]	miR-17-5p, miR-21, miR-155 and miR-196a	Association of low expression of miR-155 and miR-196a and high expression of miR-17-5P with PDAC. Correlation of high levels of miR-17-5p with metastasis and advanced PDAC.

Abbreviations: AUC, area under the curve; CP, chronic pancreatitis; HR, hazard ratio; PDAC, pancreatic ductal adenocarcinoma.

## 6. Transcriptomics

Transcriptomics involves the study of RNA transcripts. mRNA is an intermediate molecule in the process of transcription from genetic information to protein production. Hence, transcriptomics provides an instant picture of gene expression, thus it is crucial to define the orientation of cell biology and metabolism (Table 4). Several methods to explore cell transcripts have been used in the past; however, the advent of single-cell analysis has provided a much more powerful method [79]. In particular, spatial transcriptomics allows us to adequately distinguish between the different types of cells that comprise the tumor tissue and analyze the cancer architecture and the complex interactions that facilitate tumor growth and progression [80]. Indeed, five subtypes of PDAC have been distinguished through single-cell transcriptomics: basal-like A, basal-like B, hybrid, classical-A, and classical-B. Interestingly, basal-like and classical features have been shown to co-exist in some cases [10].

Moreover, the cancer subtype influences the tumor microenvironment in terms of immunosuppressive macrophages and T-cell infiltration [81]. The analysis of chemotherapy-naive and treated patients through single-cell transcriptomics with a spatial technique revealed that therapy causes further evolution in cancer cell biology. In fact, different novel subtypes were identified after treatment, and some of them were characterized by a high expression of drug-elusive genes.

As expected, changes in cancer subtypes after treatment also reflects on the tumor microenvironment composition and cell-to-cell interactions [82]. Conversely, an increase in inflammatory cytokines in the tumor microenvironment determines an adaptive response to stress in cancer cells in a paracrine modality. This mechanism further contributes to chemoresistance [83].

Inducing variation in the immune cell composition of the tumor microenvironment may be a strategy to treat chemoresistant PDAC subtypes. In a recent study by Falcomatà et al., the highly chemoresistant mesenchymal subtype was treated with a combination therapy of trametinib, an MEK inhibitor, and nintedanib, a multi-tyrosine kinase inhibitor. This approach was identified by a high-throughput drug screening. After treatment, the initial immunosuppressive microenvironment that was characterized by a predominance of TNF-expressing macrophages was replaced by an infiltration of CD8+ T-cells that reprogrammed cancer sensitivity to immune checkpoint blockade [84].

In conclusion, transcriptomics, particularly in the advanced form of spatial single-cell analysis, deepened the understanding of pancreatic cancer biology in the last few years. Despite the high costs, the still limited experience, and the need for highly experienced centers, it may provide further translational insights towards personalized cancer treatment.

**Table 4 ijms-25-07623-t004:** Main studies on transcriptomics in pancreatic cancer.

Article	Technique	Results
Moncada et al. (2020) [80]	Single-cell RNA seq	Defined subpopulations and spatial organization of cells composing PC tissues and reveal their complex interactions.
Raghavan et al. (2021) [81]	Single-cell RNA seq	Systematic profiling of metastatic PC biopsies and matched organoid models provided a view of cellular states, their regulation by tumor microenvironment, and the ability to modulate these states to impact drug responses. Cancer subtype influenced tumor microenvironment in terms of immunosuppressive macrophages and T-cell infiltration.
Hwang et al. (2022) [82]	Single-cell RNA seq + spatial transcriptomics	Identified multicellular dynamics and further evolution in PC cell biology associated with neoadjuvant treatment.
Cui Zhou et al. (2022) [83]	Single-cell RNA seq + spatial transcriptomics	Identified tumor and transitional subpopulations of cells with distinct histological features. Chemoresistance was determined by an increase in inflammatory cytokines in the tumor microenvironment as an adaptive response to stress in cancer cells.
Falcomatà et al. (2022) [84]	Single-cell RNA seq + CRISPR screen + immunophenotyping	Study of intratumor infiltration of cytotoxic and effector T-cells and sensitization of mesenchymal PC to PD-L1 immune checkpoint inhibition.
Barthel et al. (2023) [79]	Single-cell RNA seq + spatial transcriptomics	Multimodal approaches to elucidate PC biology and response to therapy.

Abbreviations: CRISPR, clustered regularly interspaced short palindromic repeats; PC, pancreatic cancer.

## 7. Radiomics

The term radiomics defines a mathematical method that extracts quantitative data and relevant and reproducible objective metrics from acquired images oriented to lesion diagnosis and prognosis, through the application of different computational models and processing techniques [85,86].

The extraction process of image-based features from lesions and surrounding tissues is based on different phases, including image reconstruction and preparation (denoising, harmonization, and segmentation), feature extraction, data integration, and modeling to predict clinical outcomes and diagnostic/prognostic factors [87]. Radiomic features can be related to the pixel/voxel distribution within the image or to different tumor behaviors, phenotypes, cancer microenvironment, genomic profiles, or specific clinical outcomes [88,89].

Recent evidence has shown that radiomics may be a useful tool for the diagnosis, staging, choice of treatments, response to treatments, and prognosis, supporting better medical decision-making for PC management [90,91,92] (Table 5).

CT is the main diagnostic tool to assess the stage of PC. For this reason, many studies in the literature have focused on the extraction of radiomic features from CT images to provide comprehensive information on the tumor’s phenotypic and textural structure [93]. For example, Park et al. demonstrated an overall accuracy of 95.2% in distinguishing autoimmune pancreatitis (AIP) from PDAC. Chu et al. obtained similar results in the assessment of CT radiomic features to differentiate PDCA from normal pancreatic tissue [94,95].

Interestingly, another radiomic study highlighted the possibility of detecting PDAC at a substantial lead time before clinical diagnosis (3–36 months) using radiomics-based machine-learning (ML) models. In a population of 155 patients and a control cohort of 265 subjects, the diagnostic performance was significantly better, with a sensitivity of 95%, a specificity of 90.3%), an AUC of 0.98, and an accuracy of 92.2%. The median time between prediagnostic CTs of the test subset and PDAC diagnosis was 386 days [96]. Similarly, an extended neural network analysis on EUS-based elastography for the differential diagnosis of PC and chronic pancreatitis showed an average testing performance of 95% on cross-validation [97].

The predicted risk of lymph node metastasis using a radiomic model was also investigated to avoid unnecessary surgery [98,99]. In terms of surgical complications after pancreatic resection, radiomic analysis was also used to predict the risk of postoperative pancreatic fistula (POPF). Zhang et al. developed and validated a radiomics-based formula for predicting POPF in patients undergoing pancreaticoduodenectomy [100].

Immunotherapy showed limited results in patients with PC. However, a radiomics-based approach to predict the expression of CD8+ tumor-infiltrating lymphocytes in patients with PDAC was used to identify candidates for immunotherapy-targeting immune checkpoint inhibitors [101].

Radiomics was also applied to predict responses to treatments and to estimate prognosis. Watson et al. tried to predict pathologic tumor responses to neoadjuvant therapy in PC. The AUC for the prediction of response to chemotherapy and resectability were 0.738 and 0.783, respectively [102]. In another study, pretreatment CT quantitative imaging biomarkers were associated with disease-free survival (DFS) and OS in patients with unresectable PC undergoing chemotherapy [103]. Similarly, Parr et al. and Cozzi et al. applied radiomic models to predict OS and locoregional recurrence of PC after stereotactic body radiation therapy, differentiating high and low risk in terms of OS [104,105].

Radiomic features were also combined with other clinical data to improve accuracy. In a large cohort of 161 patients with resected PC, CT-derived radiomic texture features of the cancer were combined with preoperative serum carbohydrate antigen 19-9 (CA 19-9) levels and the pathology Brennan score to increase prognostic accuracy [106].

Radiomics-based models, such as computer-aided diagnosis (CAD) tools, have also been proposed in several studies to improve the efficiency and accuracy of pancreatic cyst diagnosis (differential diagnosis of pancreatic cyst types, low-grade vs. high-grade IPMNs) and risk stratification in clinical workflows and treatment decision-making [107,108,109]. In particular, an EUS-based model achieved 94% accuracy in diagnosing IPMN malignancy, compared with 56% accuracy in human diagnosis. In another study using MRI to classify IPMN, Corral and colleagues achieved an AUC of 0.78 with the radiomic technique, which was comparable to human performance according to the American Gastroenterology Association guidelines (0.76) and the Fukuoka criteria (0.77) [110,111].

Moreover, Koay et al. demonstrated that the quantitative analysis of the tumor–stromal interface was a significant prognostic indicator in three different cohorts of patients with PDAC treated with surgery or chemotherapy [112]. Similarly, Chakraborty et al. and Attiyeh et al. showed that their analysis could be a promising tool to quantify heterogeneity in CT images, predicting survival in patients with PDAC and elaborating survival prediction models because of their association with OS [106,113]. These data were not confirmed by Cassinotto et al., who found that there was no association between the texture signature and DFS. However, they showed a shorter DFS, higher tumor grade, greater aggressiveness, and lymph node invasion in tumors with hypoattenuation on the portal-venous phase of CT scans [114].

Furthermore, the combination of radiomics data with genomic markers, such as micro-RNA expression, may offer the most appropriate therapeutic choice in the perspective of increasingly personalized precision medicine. In this context, Iwatate et al. investigated the prediction of p53 and PD-L1 expression from CT images, demonstrating a statistically significant association between radiogenomic-predicted p53 mutations and poor prognosis [115].

**Table 5 ijms-25-07623-t005:** Main studies on radiomics in pancreatic cancer.

Article	Imaging	Results
Săftoiu et al. (2008) [97]	EUS	Sensitivity of 91.4%, specificity of 87.9%, and accuracy of 89.7% in differentiating benign (normal pancreas, CP) and malignant masses (PDAC, NET), respectively. Positive predictive value of 88.9% and negative predictive value of 90.6%.
Chakraborty et al. (2017) [113]	CT	Texture analysis to quantify heterogeneity in CT images to accurately predict 2-year survival in patients with PDAC (AUC of 0.90 and accuracy of 82.86% with the leave-one-image-out technique and an AUC of 0.80 and accuracy of 75.0% with three-fold cross-validation).
Cassinotto et al. (2017) [114]	CT	Lymph node ratio (R2 = 0.15), kurtosis (R2 = 0.08), and CENTRAL-AV (R2 = 0.04) associated with early-recurrence in resectable PDAC. CENTRAL-AV < 62 Hounsfield units associated with a shorter 1-year DFS (35% versus 68%, *p* = 0.004).
Zhang et al. (2018) [100]	CT	Rad score could predict postoperative pancreatic fistula in patients undergoing pancreaticoduodenectomy with an AUC of 0.82 in the training cohort and of 0.76 in the validation cohort.
Attiyeh et al. (2018) [106]	CT	Quantitative image features combined with CA 19-9 achieved a c-index of 0.69 [integrated Brier score (IBS) 0.224] on the test data, while combining CA 19-9, imaging, and the Brennan score achieved a c-index of 0.74 (IBS 0.200) on the test data in resected PDAC.
Chu et al. (2019) [95]	CT	Overall accuracy of 99.2% and AUC 99.9% for random forest binary classification (PDAC and normal pancreas). 100% of PDAC correctly classified with 100% sensitivity and 98.5% specificity.
Bian et al. (2019) [98]	CT	Significant association between the arterial rad-score and the lymph node metastasis (*p* < 0.0001) in PDAC.
Cozzi et al. (2019) [105]	CT	Significant association of clinical-radiomic signature with overall survival in training and validation sets (*p* = 0.01 and 0.05; concordance index 0.73 and 0.75 respectively) after stereotactic body radiotherapy for PDAC.
Wei et al. (2019) [107]	CT	Radiomics-based computer-aided diagnosis scheme could increase preoperative diagnostic accuracy (AUC = 0.767, sensitivity = 0.686, specificity = 0.709) in differentiating pancreatic serous cystic neoplasms from other pancreatic cystic neoplasms.
Corral et al. (2019) [110]	MRI	Deep learning protocol with high sensitivity and specificity to detect dysplasia (92% and 52%, respectively), high-grade dysplasia or cancer (75% and 78%, respectively), and an accuracy comparable to radiologic criteria (AUC 0.76 for American Gastroenterology Association, 0.77 for Fukuoka and 0.78 for the deep learning protocol, *p* = 0.90).
Kuwahara et al. (2019) [111]	EUS	Artificial intelligence deep learning algorithm with a significantly greater score for malignant IPMNs than benign IPMNs (0.808 vs. 0.104, *p* < 0.001). High sensitivity, specificity and accuracy of AI malignant probability (95.7%, 92.6% and 94.0%, respectively) in detecting malignant IPMNs.
Liu et al. (2020) [99]	CT	Radiomics LOG model with higher predictive efficiency compared to the conventional preoperative evaluation method of lymph node status (AUC = 0.84; 95% CI, 0.758~0.925 vs. AUC = 0.68; 95% CI, 0.566~0.798) in the resectable PDAC.
Park et al. (2020) [94]	CT	Differentiation of AIP from PDAC with 95.2% accuracy (59/62; 95% CI, 89.8–100%) and AUC of 0.975 (95% CI, 0.936–1.0). 100% of PDAC correctly classified with thin-slice venous phase with 89.7% sensitivity (26/29; 95% CI, 78.6–100%) and 100% specificity (33/33; 95% CI, 93–100%).
Parr et al. (2020) [104]	CT	Role of a 6-feature radiomic signature in achieving better overall survival prediction performance (mean concordance index 0.66 vs. 0.54) and of a 7-feature radiomic signature in predicting recurrence (mean concordance index 0.78 vs. 0.66 on resampled cross-validation test sets) in PDAC.
Li et al. (2021) [101]	CT	Extreme gradient boosting classifier (XGBoost) showed sensitivity, specificity, accuracy, positive and negative predictive values of 0.81, 0.60, 0.69, 0.63, and 0.79, respectively, for the training set, and 0.81, 0.58, 0.68, 0.61, and 0.79, respectively, for the validation set in predicting CD8+ tumor-infiltrating lymphocyte expression levels in PDAC.
Huang et al. (2021) [108]	CT	Arterial radiomics model constructed by the 3D-ROI feature performed better (AUC: 0.914) than venous (AUC: 0.815) in predicting the invasiveness of pancreatic solid pseudopapillary neoplasms.
Watson et al. (2021) [102]		Deep learning models predicted pathologic tumor response to neoadjuvant therapy in PC (AUC for the response to chemotherapy and resectability were 0.738 and 0.783, respectively).
Watson et al. (2021) [109]	CT	Deep learning model correctly predicted malignancy of pancreatic cystic neoplasms (3 of 3 malignant lesions and 5 of 6 benign lesions), performing better than consensus guidelines (2 of 3 malignant lesions as high risk and 4 of 6 benign lesions as worrisome).
Mukherjee et al. (2022) [96]	CT	Support random machine with the highest sensitivity (95.5%; 95% CI, 85.5–100.0), specificity (90.3; 95% CI, 84.3–91.5), F1-score (89.5; 95% CI, 82.3–91.7), AUC (0.98; 95% CI, 0.94–0.98) and accuracy (92.2%; 95% CI, 86.7–93.7) for differentiation of PDAC at the prediagnostic stage from normal pancreas.

Abbreviations: AIP, autoimmune pancreatitis; AUC, area under the curve; CP, chronic pancreatitis; CT, computed tomography; EUS, endoscopic ultrasound; IPMN, intraductal papillary mucinous neoplasia; MRI, magnetic resonance imaging; NET, neuroendocrine tumor; PC, pancreatic cancer; PDAC, pancreatic ductal adenocarcinoma.

## 8. Single-Cell Profiling, Multi-Omics, Future Applications and Perspectives of Omics Sciences

Single-cell profiling is a powerful emerging technique for investigating the complex interplay between the different cells that comprise the tumor microenvironment. Multiple omics can be explored by this technology [79]. Indeed, several different recent studies explored the immune microenvironment of PDAC, combining single-cell RNA transcriptomics, metabolomics, proteomics, and multicolor immunohistochemistry [116,117,118,119]. However, the combination of several advanced molecular techniques creates further interpretation issues that need to be addressed.

Another intriguing aspect to develop is the ability to achieve a non-invasive diagnosis and prognosis using biological fluids, the so-called liquid biopsy. This term encompasses the sampling and analysis of circulating tumor cells (CTCs), EVs, and tumor DNA (ctDNA) from body fluids. Each of these analyses has shown potential benefits and disadvantages [120]. In a recent meta-analysis of several studies on CTCs, the overall detection rate was 65%. The positivity for CTCs in the blood was a negative prognostic factor and was more frequent in advanced disease. The detection was higher in portal vein samples, whose invasiveness was clearly higher compared with peripheral vein sampling [121]. The study of EVs requires a deeper understanding of cancer biology. Indeed, all cells release EVs in the blood and other biological fluids. The ability to attribute the EVs to the cell of origin depends on the identification of biomarkers that are specific to the generating cell [122]. As for PC, Melo et al. demonstrated that patients with PDAC had significantly higher serum levels of glypican-1-positive (GPC-1) exosomes compared with patients with chronic pancreatitis and healthy controls (*p* < 0.0001) [123]. However, other studies have also distinguished patients with malignant and benign disease [124,125]. In all patients with PDAC (n = 190), GPC-1+ exosomes contained mutant KRAS. Hence, EVs may be an adequate substrate for the non-invasive omics analysis of PC. In fact, several proteomic and genomic studies have been conducted using EVs [75,123]. However, among liquid biopsy approaches, EV research seems to be the most promising, yet it is still far from being applicable in clinical practice [122]. Indeed, despite the efforts of international societies, there is still a need for further standardization and validation of EV isolation techniques [126,127].

The idea to develop cancer cell cultures to test anticancer therapies dates back to 1950s. However, monolayer cell cultures are poor predictors of in vivo tumor biology, thus 3D cell cultures have been developed [128]. To overcome these limitations, cell cultures that include growth factors, stromal elements, or scaffold supports were developed to sustain the growth of the cancer cells [129]. A group of cells growing in a 3D structure that is generated directly from primary tissues is called an organoid. Organoids have the ability to self-renew and self-organize, and they can reproduce the histology and function of the tissue of origin [130]. PDAC organoids can be generated either from surgical samples or fine-needle aspiration (FNA) with a remarkable success rate [131].

Since the first demonstration, several methods to harbor human PDAC organoids have been developed and they have also been used to identify different disease subtypes [132]. Pharmacogenomics seems to be the most important field for the application of organoids. In fact, they may be used to test anticancer drugs or their associations with the specific tumor of the patient in order to predict responses and avoid ineffective treatments with significant side effects. This approach may guide clinical management using a personalized drug approach [133,134,135] (Figure 1).

The microbiota hosts trillions of microbes that interplay a complex interaction with the host and can be a driver for carcinogenesis [136]. Several recent studies have revealed that it may play a role in the development and progression of cancer and in the response to treatments [137].

In a recent study using shotgun metagenomic sequencing, an abundance in *Veillonella atypica*, *Fusobacterium nucleatum/hwasookii*, and *Alloscardovia omnicolens,* and a depletion in *Romboutsia timonensis*, *Faecalibacterium prausnitzii*, *Bacteroides coprocola*, and *Bifidobacterium bifidum* was observed in the gut microbiota of patients with PC when compared with patients with PC and healthy controls. This microbiota fingerprint was used as a diagnostic tool in a validation population of patients with PC with an area under the receiver operating characteristic curve (AUROC) of 0.84. The diagnostic performance reached 0.94 when the microbiota alteration was combined with the use of CA 19-9. The specific alteration of microbiota in patients with PC may be a diagnostic tool [138].

Moreover, microbiota-derived molecules can influence the efficacy of chemotherapy. The microbiota-derived tryptophan metabolite indole-3-acetic acid (3-IAA) was increased in responders to chemotherapy [139]. Interestingly, high levels of this molecule can be toxic to cells with a high concentration of myeloperoxidase (MPO), such as neutrophils [140]. Neutrophils are highly abundant in PDAC and are associated with poorer prognosis [141].

Moreover, the promising results of fecal microbiota transplantation in other gastrointestinal diseases may offer a possible adjuvant therapy, particularly for the prevention of chemotherapy intolerance and immune modulation [142]. Hence, the study of the microbiota may help to individualize PC care in the future [143].

Finally, the recent development of artificial intelligence (AI) systems may further enhance the application of omics sciences for the study and management of PC. Indeed, deciphering the complex amount of information from combined multi-omics sciences may be impossible for conventional data analysis. This approach was recently applied to predict pancreatic cancer prognostic outcomes by combining advanced multi-omic features from patients’ datasets [144]. Hence, AI may play a role in each field of research for pancreatic cancer, such as early identification of the disease, discovery of new biomarkers, combination of information from different omics techniques to understand its biology, stratification of the risk of progression, and development of new strategies for treatment [145].

## 9. Conclusions

PC is still one of the major health problems due to its late diagnosis, inapplicability for surgical treatment, and inefficacy of oncological treatments for advanced disease in most cases.

Conventional research has not significantly improved pancreatic cancer care during the last decades. Hence, research trends have taken the modern but complex approach of omics sciences. Although ongoing efforts to turn omics research into clinical practice is not always feasible, these preliminary evidence will help to unravel the complex conundrum of PC.

The integration of information derived from different omics sciences remains a topic to address in the near future. AI may be a solution for analyzing the vast body of molecular information acquired through omics techniques and translating it to clinical practice. The hope driving this complex field is that it may finally result in a more effective approach for the study and clinical management of PC.

## Figures and Tables

**Figure 1 ijms-25-07623-f001:**
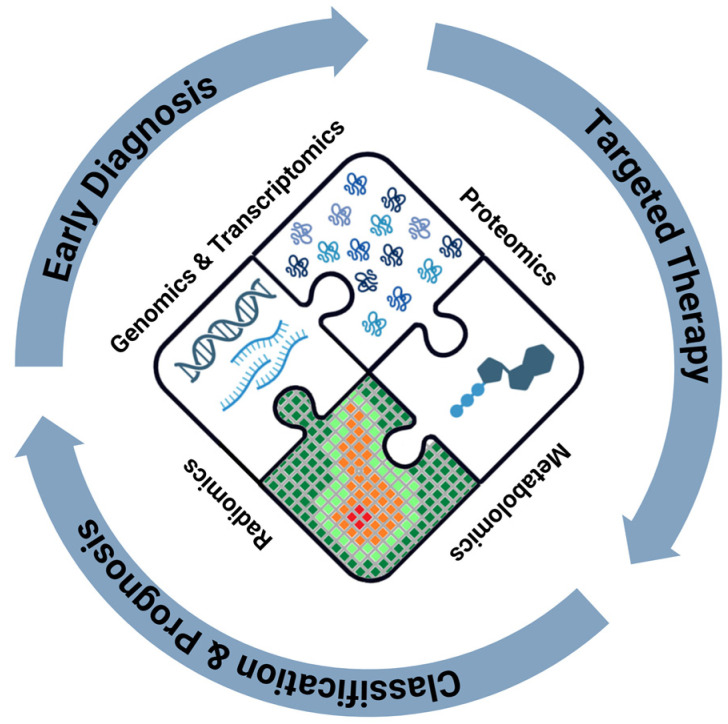
Aims of integrated omics approach to pancreatic cancer care. Created with BioRender.com.

## Data Availability

No new data were created or analyzed in this study. Data sharing was not applicable to this article.

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
