# Peer review of "Understanding the Conundrum of Pancreatic Cancer in the Omics Sciences Era"

_ijms, 2024, doi:10.3390/ijms25147623_

Round 1
Reviewer 1 Report
Comments and Suggestions for Authors
The authors here provided a comprehensive summary of multi-omics-based dignosis and prognosis of pancreatic cancer (PC). The contents about the omic applications in this manuscript mainly includes metabolic changes, proteomics changes, genomics mutations, and radiomic changes. After my careful reading, I find the manuscript should be further modified to improve the readability.
The present form is logically confusing, and even the core ideas can not be easily grasped by the readers. Especically, no any figures or tables are provided to help readers to understand the ideas. Maybe this present form will be hard to stimulate the deep thought by the majority in this field. It is better for the authors to resubmit their manuscript for the next considerations.
Here are some compulsory modifications for the reference.
1. In each part, a table should be provided to list all possible biomarkers indentified by multi-omics.
2. In each part, an illustrative figrue should be added to dissect the mode of action of some key proteins or genes or metabolites.
3. A separate part about the combinations of multiple omic technologies in the applications of PC should be added to strengthen the thought.
4. I think a single-cell omic application should be also discussed, and analyze its difference from the bulk-sample based omic data.
5. RNA-seq data reveals the transcriptional dynamics in PC progression. Transcriptome analysis is still indispensible in the discussion.
6. I suggest the authors specially find some key biomarkers to present the PC progression with a Figure.
7. Maybe the main contents of omic applications can be better to be subdivided into the following parts in sequence: phenome (including radiomics), genome, transcriptome, proteome, metabolome, single-cell omics, multi-omics.
Author Response
We thank you for your attention, constructive comments and thorough work to identify areas of our manuscript that needed improvements.
In the following section, we addressed the specific points of your revision and described changes we have made in the text. Moreover, we revised all the sections to fit your indications. Changes have been made in “revision mode”.
- In each part, a table should be provided to list all possible biomarkers indentified by multi-omics.
According to your suggestion, we added a table to list the most promising biomarkers for each section of the review.
- In each part, an illustrative figrue should be added to dissect the mode of action of some key proteins or genes or metabolites.
We preferred to add a figure to explain the aims of the integrated omics approach to pancreatic cancer care.
- A separate part about the combinations of multiple omic technologies in the applications of PC should be added to strengthen the thought.
We modified the last section of the review to include the multi-omics approach.
- I think a single-cell omic application should be also discussed, and analyze its difference from the bulk-sample based omic data.
We included a discussion about the single-cell omic approach in the last section of the review.
- RNA-seq data reveals the transcriptional dynamics in PC progression. Transcriptome analysis is still indispensible in the discussion.
According to your suggestion, we added a whole section on trascriptomics in the review.
- I suggest the authors specially find some key biomarkers to present the PC progression with a Figure.
We presented key biomarkers in tables in each section, since they would easier summarize the main evidence.
- Maybe the main contents of omic applications can be better to be subdivided into the following parts in sequence: phenome (including radiomics), genome, transcriptome, proteome, metabolome, single-cell omics, multi-omics.
We endorsed your suggested partition of the sections of the review. In particular, we added a section on metabolomics and a section on transcriptomics. However, we preferred to discuss single-cell omics and multi-omics together in section 8.
Reviewer 2 Report
Comments and Suggestions for Authors
The review "Understanding the conundrum of pancreatic cancer in the omics sciences era" presents modern information about pancreatic cancer. I would like to make the following recommendations:
1. The introduction chapter is well-executed.
2. Add a Materials and Methods chapter.
3. In Chapter 5, discuss the importance of ERCP in the diagnosis of pancreatic cancer and the risks associated with this procedure – more information can be found at https://doi.org/10.3390/jpm13091356.
4. In the future perspectives chapter, add information about the influence of microbiota transfer on pancreatic cancer. More information can be found at: https://doi.org/10.3390/diagnostics14090861.
5. Add a table summarizing the main articles in the literature with a similar theme and the results presented by them.
Author Response
We thank you for your attention, constructive comments and thorough work to identify areas of our manuscript that needed improvements.
In the following section, we addressed the specific points of your revision and described changes we have made in the text. Moreover, we revised all the sections to fit your indications. Changes have been made in “revision mode”.
- The introduction chapter is well-executed.
We appreciated that you found this section well-written.
- Add a Materials and Methods chapter.
We thank you for the suggestion. However, this is not a systematic review, but a narrative one. Hence, we omitted this section, according to journal guidelines.
- In Chapter 5, discuss the importance of ERCP in the diagnosis of pancreatic cancer and the risks associated with this procedure – more information can be found at https://doi.org/10.3390/jpm13091356.
Unfortunately, the article you suggested do not fit the purpose of our review. However, we are working on another review on the complications of pancreatic cancer and we could cite it there.
- In the future perspectives chapter, add information about the influence of microbiota transfer on pancreatic cancer. More information can be found at: https://doi.org/10.3390/diagnostics14090861.
We added a sentence on the modulation of the microbiota through fecal microbiota transplantation and cited the suggested article.
- Add a table summarizing the main articles in the literature with a similar theme and the results presented by them.
According to your suggestion, we presented key biomarkers in a table for each section.
Round 2
Reviewer 1 Report
Comments and Suggestions for Authors
The authors have well addressed all my concerns. It can be accepted for publication in its present form.
Reviewer 2 Report
Comments and Suggestions for Authors
The authors have made all the recommended changes.